# Opioid Antagonist in the Treatment of Ischemic Stroke

**DOI:** 10.3390/brainsci11060805

**Published:** 2021-06-18

**Authors:** Juan Fernando Ortiz, Claudio Cruz, Amrapali Patel, Mahika Khurana, Ahmed Eissa-Garcés, Ivan Mateo Alzamora, Taras Halan, Abbas Altamimi, Samir Ruxmohan, Urvish K. Patel

**Affiliations:** 1School of Medicine, Colegio de Ciencias de la Salud, Universidad San Francisco de Quito, Quito 170901, Ecuador; cruzclaudio9014@gmail.com (C.C.); aeissag@estud.usfq.edu.ec (A.E.-G.); mateoalzamora@hotmail.com (I.M.A.); 2Neurology Department, School of Medicine, AMC MET Medical College, Ahmedabad 380008, India; 3Public Health Department, University of California Berkeley, Berkeley, CA 94720, USA; mahikakhurana@gmail.com; 4School of Medicine, Ternopil National Medical University, 46002 Ternopil, Ukraine; trshalan@gmail.com; 5Emergency Department, Amiri Hospital, Sharq 15300, Kuwait; dr.abbasaltamimi@gmail.com; 6Department of Neurology, Larkin Community Hospital, Miami, FL 10029, USA; ruxmohan@yahoo.com; 7Public Health Department, Icahn School of Medicine at Mount Sinai, New York, NY 10029, USA; dr.urvish.patel@gmail.com

**Keywords:** stroke, naloxone, anti-opioids, nalmefene, clinical trials

## Abstract

Stroke is a leading cause of death and disability, and novel treatments need to be found, particularly drugs with neuroprotective and restorative effects. Lately, there has been an increased interest in the relationship between opioids and ischemic stroke. To further appreciate this association between opioids and stroke, we conducted a systematic review to investigate anti-opioid medication’s effectiveness in treating ischemic stroke. We used PubMed advanced-strategy and Google Scholar searches and only included full-text clinical trials on humans and written in the English language. After applying the inclusion/exclusion criteria, seven clinical trials were reviewed. Only one of the naloxone and nalmefene clinical trials showed statistically favorable results. Overall, the nalmefene clinical trials used more updated measures (NIHSS, GOS) to evaluate recovery and functional status in ischemic stroke patients than the naloxone clinical trials. There was less bias in the nalmefene clinical trials. Animal and in vitro studies have showed promising results. Additional research should be conducted with new clinical trials of both drugs with larger samples in patients less than 70 years old and moderate to severe infarcts.

## 1. Introduction

Stroke is the fifth leading cause of death in the United States [1]; however, stroke management has not changed substantially since the introduction of tissue plasmin activator (tPA). Currently, tPA is the only indicated drug to treat an ischemic stroke, and only 3% of patients are eligible for treatment with tPA [1]. Moreover, tPA does not exhibit neuroprotective or restorative effects. Novel treatments need to be found, particularly drugs with neuroprotective and restorative effects.

Endogenous opioids may be released after an ischemic or traumatic injury, worsening the initial insult [2]. It has been reported that opioid users have an increased risk of developing ischemic stroke [3]. According to Lee et al., cancer patients treated with opioids have a 12% greater risk of developing stroke than patients not treated with opioids [3]. In another case-controlled study, Moqaddam et al. found that opioid dependency was an independent risk factor for a stroke, with an odds ratio of 2.6 [4]. Also, opioids were found to increase the risk of atrial fibrillation, which is an independent factor for ischemic stroke [5]. If opioids are linked to ischemic stroke, it is plausible that opioid antagonists may have a protective effect on stroke patients. There is evidence that naloxone can reverse ischemic symptoms, and at high concentrations, even having an antioxidant effect, but the information is limited [6,7].

We conducted a systematic review of clinical trials of anti-opioid medications for the treatment of ischemic stroke. Only clinical trials conducted on human subjects and published in the English language were analyzed. We investigated the drugs’ clinical trial efficacies and explored possible anti-opioid medication mechanisms to treat ischemic stroke.

## 2. Materials and Methods

### 2.1. Protocol

We carried out a systematic review using the preferred reporting items for systematic reviews (PRISMA) and meta-analysis [7].

#### 2.1.1. Eligibility Criteria and Study Selection

We only included clinical trials conducted on humans and written in English. Animal studies were excluded. We excluded papers that did not fulfill the aims of our study. After screening the studies, we only included papers with one of the following characteristics: (1) Patients: individuals with ischemic stroke and above 18 years old; (2) Intervention: use of an anti-opioid medication in patients with ischemic stroke; (3) Comparator: placebo or control group; and (4) Outcomes: any neurological outcome to evaluate functional outcome or recovery after a stroke.

#### 2.1.2. Database and Search Strategy

We used the PubMed database for this systematic review. The search was conducted between 1 March 2021 and 15 March 2021. We used an advanced search strategy with the following terms: (stroke [Title/Abstract]) AND (naloxone [Title/Abstract]).

#### 2.1.3. Data Extraction and Analysis

We collected the following information from each paper the methods, such as dose, duration, and route of administration and number of participants, study design, and patients selection. We also extracted the main results, including the outcome measures and main limitations in each clinical trials. We analyzed the studies’ primary and secondary goals and gathered the main conclusions from each study.

#### 2.1.4. Bias Assessment

We used the Cochrane collaboration risk-of-bias tool to assess the bias encountered in each study [8].

## 3. Results

Figure 1 shows the results of the study using a PRISMA Flow chart.

We found four, three, and zero clinical trials discussing naloxone, nalmefene, and naltrexone, respectively.

### 3.1. Study Characteristics

We found four clinical trials that specifically discussed the role of naloxone in the acute setting of ischemic stroke. Table 1 presents the main characteristics of each study.

Table 2 shows the outcomes and conclusions of the clinical trials of this systematic review.

Among the studies, only one clinical trial reported statistically significant results for naloxone [2]; while in the clinical trials of nalmefene, only one clinical trial was statistically significant as well [13]. Because all studies used different parameters, it was not possible to do a statistical analysis.

### 3.2. Study Limitations

The studies by Jabaily et al. and Fallis et al. were incorrectly randomized with uneven distribution in the treatment and control group [6,10]. The Jabaily et al.’s study was single-blinded, making it very susceptible to observer bias [10]. Czlonkowske et al.’s study was a double-blind clinical study with a small sample. The authors’ argued for the study’s internal and external validity because the randomization was not optimal [2]. Federico’s study was a pilot study. Again, the study had a small sample size that affected the validity, and the author argued that there was no justification for studies on larger scales [5].

Clark et al.’s first study model was limited because it included patient subset analyses, which increases the chance of type II errors [11]. Clark et al.’s second study had the same limitation and also included too few samples for adequate study power [12]. The placebo group’s high recovery rate led to a greater chance of type II errors and confounding baseline differences [12].

Finally, there were some limitations in Li et al.’s study [13]. The sample was small, the detection of serum Matrix metallopeptidase 9 MMP-9 level and the magnetic resonance imaging (MRI) perfusion head imaging was not done for all participants, and the time-frame of the study was only 20 days [13].

Overall, the nalmefene clinical trials exhibited less bias compared to the naloxone clinical trials. Figure 2 summarizes the main biases found in all studies.

## 4. Discussion

### 4.1. Naloxone Clinical Trials

At the doses administered in Jabaily’s study, naloxone does not exhibit important systemic effects; however, the author stated that different doses of naloxone could give different results [10]. Overall, the study did not show any significant results, although three patients had improved neurological symptoms. Rather than a significant result, the author concluded that the improvement of neurological symptoms using naloxone is rare. The study size was too small [10].

In the study by Fallis et al., there was no significant difference between the groups [6]. Moreover, the score used to evaluate the patients was unusual, not practical nowadays. Four points were analyzed to evaluate the neurological score: maximal strength, facial movement, speech, and consciousness level. Another study suggested that naloxone decreased blood pressure [14]; however, blood pressure was not significantly changed in the study. Initially, 0.4 mg was administered, equivalent to the dose used to reverse a morphine overdose; this was later increased to 4 mg because no results were observed. Neither group showed any significant differences [6]. One critique of this study, besides the small sample size, was the antiquated parameters used to evaluate the patients [6].

In Federico et al.’s study, traditional scales were used to evaluate the patients, which included the Barthel Index and the Canadian Neurological Scale [9]. Both study groups had similar characteristics, which reduced the selection bias. Again, no significant differences were found between the groups, and it was suggested that naloxone might only be helpful during the early stages of an ischemic stroke. κ-opioid receptors seem to have a more prominent role in cerebral ischemia compared with μ opioid receptors. Naloxone has more affinity for μ-opioid receptors compared to the anti-opioid nalmefene [15]. Interestingly, hyperkinetic movements were momentarily seen in some patients.

The results of Czlonkowke et al.’s study differed from the other studies because they found significant differences in the recovery time of patients treated with naloxone [2]. A dose as low as 0.8 mg produced a significant reduction in neurological deficits that lasted up to 2 weeks. However, the study emphasized that the groups’ randomization was not optimal, and type II errors may have occurred [2].

### 4.2. Nalmefene Clinical Trials

Two studies conducted by Clark et al. showed that 60 mg of nalmefene could be safely given within 24 h [11,12]. However, the patients treated with nalmefene reported more side effects, specifically nausea and vomiting. The study found no statistical difference in any outcome [11]. However, an analysis of patients younger than 70 years showed a positive trend in the treatment group’s outcomes compared to the controls. However, the authors emphasized that the results could have resulted from a type II error. The authors also suggested that the results should be replicated to confirm the findings [11].

A subsequent analysis by the same authors in 2000 produced similar results [12]; it also found a favorable tendency for patients younger than 70 years, but the results were not statistically significant. The authors argued that studies should focus on patients with moderate to severe infarcts. They suggested that clinical trials should be undertaken on patients with moderate to severe infarcts with NIHSSs > 8. Finally, according to the authors, there is a need to study efficacy in patients co-treated with thrombolytics within 3 h of a stroke [12].

Li et al.’s study was the most recent clinical trial. The study showed that nalmefene improved injured brain tissue by antagonizing κ opioid receptors [13]. In contrast to the naloxone clinical trials, Li et al.’s study used conventional, modern measures to evaluate functional outcomes and recovery in patients with ischemic strokes. In the treatment group, the NIHSS was decreased, the Glasgow Coma Scale was increased, matrix metalloproteinase-9 was decreased, and magnetic resonance imaging perfusion was increased compared to the treatment group. It is important to note that this study has only released preliminary results, and the final results are pending.

### 4.3. In Vitro Studies and Aniaml Stuies

While the naloxone clinical trials reviewed here are old, there are recent in vitro and animal studies that show that naloxone is effective.

In a study by Wang et al., rats were subject to middle cerebral artery occlusion, permanent occlusion of the middle cerebral artery, and received 0.5–1.2 mg/kg IV of naloxone. There was an evaluation of ischemia after 24 h using the McGraw Stroke Index, and then the rats were killed and brain tissue was collected for study purposes [16].

The study showed that naloxone in a dose-dependent fashion decreased NF-κB p65 in the ischemic penumbra, diminishing the apoptotic mitochondrial pathway by increasing BCL-2 levels and decreasing Bax levels, which decreased caspase-3 and 9 activations by inhibiting the release of cytochrome c [16].

Neuroinflammation plays an important role in the activation. A study from 2018 Anttila et al., showed that intranasal use of naloxone decreased microglia/macrophage activation, which improved behavioral recovery and cerebral ischemia. The study was conducted on rats and investigated the activation of microglia/macrophage in the astrocyte cells over the peri-infarct area and neuronal death in the ipsilateral thalamus. CD68 (+) was studied because it is a marker of microglia/macrophage and was diminished at the end of the study. The study was the first of its kind showing inhibition of TNF alpha secretion from stroke-activated microglia/macrophages after naloxone. It supports the role of naloxone involving its positive effects on stroke via TLR signaling and depleting oxidative stress [17].

The study warrants a need for a future investigation into the role of thalamic injury in cortical stroke and additional clinical studies to suggest varying dosing patterns to enhance the post-stroke naloxone dosing regimen [18].

Oxidative stress is a major component of motor neuron degenerative disease. In the Hsu et al. study, using a distinguished in-vitro model, the role of nanomolar naloxone was investigated as a neural protection agent under oxidative stress (H_2_O_2_) in cells with survival Motor Neuron (Smn) protein deficiency in NSC34, a mouse neuroblastoma-spinal cord motor neuron-like cell line [18].

The study determined that naloxone at nanomolar concentrations was effective against H2O2-induced neurotoxicity. Neurotoxicity was avoided by diminishing the apoptotic pathway and inducing Smn-dependent Bcl-2 expression. Naloxone decreased apoptotic death by reducing cytochrome c release from the mitochondria to the cytosol, which increased the ratio of Bcl-2 to Bax, attenuating caspase-dependent signaling [18].

Nanomolar naloxone alone improved Smn and Bcl-2 expression in NSC34 cells and NSC34 cells with Smn deficiency [18]. At a higher concentration, naloxone provided neuroprotection against H_2_O_2_-induced cytotoxicity in Smn knockdown NSC34 cells. Hence, the study explored the involvement of Smn protein in neurotoxicity and neuroprotection, proposing a direction in finding potential candidates for neuroprotection [19].

Regarding Naltrone, a study conducted on mice investigates microgliosis in the CA1 region in the hippocampus as a result of neurological death and the role of Toll-like receptor 4 (TLR-4). Naltrexone showed to be protective by blockage of the TLR4, which decreased microglial activation and infiltration of T and B cells to the site of inflammation [19]. IL-10, an anti-inflammatory cytokine, was unaffected by naltrexone, which suggests an alternative pathway for microglial activation, which was demonstrated by increasing the expression of Arginase-1. The study suggests that naltrexone and other TLR4 antagonists could strategically improve memory and executive function after cerebral ischemia in the hippocampus [19].

### 4.4. New Directions for the Anti-Opioid Medication Treatment of Ischemic Stroke

The naloxone clinical trials were 29–37 years old [2,6,9,10]. Moreover, the clinical trials used antiquated variables to evaluate stroke recovery [2,6,9,10]. Three out of four clinical trials recommended that larger clinical trials be undertaken; however, Czlonkowska’s study did show positive results [2]. These studies are old, have small samples, and had antiquated methods, so the lack of results should not be a reason not to undertake further clinical trials.

Naloxone appears to modulate calcium flow, stabilize lysosomal membranes, and decrease N-Methyl-D-aspartate toxicity (NMDA) [7].

The molecular mechanisms of ischemic stroke are complex [20]. The ischemic core cells die mostly through necrosis, while apoptosis in the ischemic penumbra (transition zone) is the predominant mechanism [20]. As mentioned before, naloxone decreases the activations of caspases, a mechanism of apoptosis [16]. So, in theory, naloxone would be beneficial for cells in the penumbra or transitional zone. However, the penumbra is complex, and other mechanisms also seem to be involved, such as necrosis, necroptosis, or autophagy [20].

κ opioid receptors are upregulated, and endogenous opioids, such as dynorphin, are released during cerebral ischemia. Unsuccessful naloxone clinical trials led to the study of nalmefene, which antagonizes mainly κ receptors. Nalmefene has the particularity that it antagonizes κ opioid receptors substantially more than naloxone [12].

Additionally, nalmefene seems to antagonize NMDA receptors, which are released during the initial stages of an ischemic stroke. [12]. During an ischemic stroke’s primary events, vessel occlusion leads to decreased ATP and glucose utilization, which generates Na+, Ca2+ influx, and K+ efflux. Then, there is edema and membrane disruption, which generate glutamate release, activating NMDA and AMPA receptors [20]. Nalmefene antagonizes NMDA receptors, which in theory would indicate that the drug is effective during the early stages of ischemic strokes [12].

Although Clark et al.’s clinical trials did not reach statistical significance, they indicated a favorable tendency for patients aged less than 70 [12]. We did not find a good explanation for these findings. However, ischemic strokes in young patients are rising and now constitute 15–18% of all ischemic strokes [21]. So, new studies should also investigate the mechanism of stroke, and also there should be a focus on how the mechanism of ischemic stroke may change with age.

Although still preliminary, the data by Li et al. is promising because it exhibits statistical significance with the outcomes described above. [13]. Nalmefene was evaluated for ischemic recovery in patients with infarcts in the middle cerebral artery (large vessel stroke). This study’s advantage was the use of adequate and modern scales to evaluate functional outcome and recovery in patients with ischemic stroke. It was also concluded in Clark’s study suggest that patients with large vessel stroke and NIHSS > 8 might benefit more from therapy [12]. The study of Li et al. was conducted only in patients with large vessel infarcts, explaining the favorable results.

## 5. Conclusions

The efficacy of naloxone, a μ receptor antagonist, for ischemic stroke is still debatable. Only one clinical trial showed statistically significant results. Clinical trials of nalfepeme, a κ opioid receptor antagonist, showed more promising results than the naloxone clinical trials. However, the clinical trials of nalmefene used more updated measures and extensive samples. Only one clinical trial with nalmefene showed positive results.

The naloxone clinical trials have antiquated scales to evaluate the patients, so future studies with updated scales could bring positive results. In general, there was less bias in the nalmefene clinical trials compared to the naloxone studies.

More clinical trials of both drugs need to be conducted to establish their efficacy. New clinical trials will benefit from larger samples, NIHSS > 8, and large vessels. This would lead to studies having a bigger statistical power to appreciate a difference. Knowing the pathological mechanism of stroke will favor predicting if a drug will be efficient for the treatment and should be a cornerstone for future studies.

## Figures and Tables

**Figure 1 brainsci-11-00805-f001:**
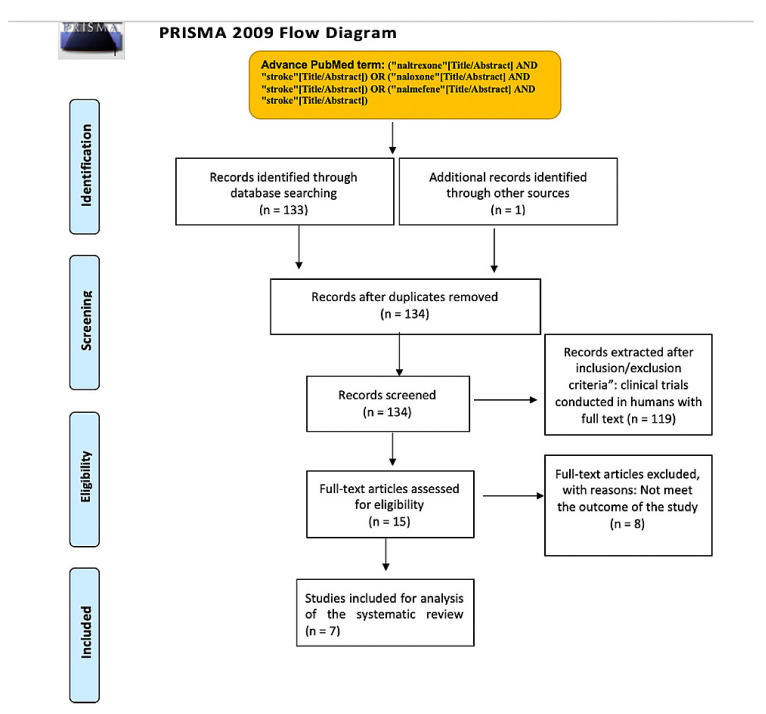
PRISMA Flowchart of the systematic review.

**Figure 2 brainsci-11-00805-f002:**
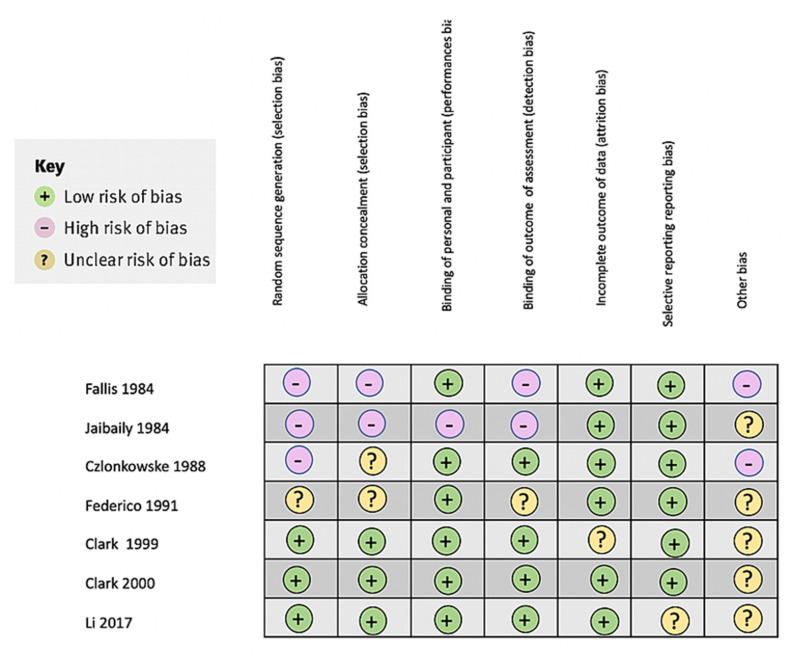
Bias analysis of the systematic review of the study.

**Table 1 brainsci-11-00805-t001:** Characteristics of clinical trials.

Author and Year of Publication	Country	Study Design	No. of pts. in the Treatment Group	No. of pts. in the Control Group	Patient Selection	Dose, Duration, Route, of Administration
Fallis et al. (1984) [6]	United States	Double-blind clinical trial	15	0	Stroke patients with deficits for 8–60 h.	Naloxone injection with normal saline. The first three patients received 0.4 mg, and the remaining patients received 4.0 mg.
Federico et al. (1991) [9]	Italy	Double- blind clinical trial, pilot study	12	12	Subjects under 80 years old, within 12 h of the onset of symptoms, not in coma, and having a negative CT scan for a previous hemorrhage, ischemic, and infarct.	Naloxone 5 mL/kg diluted in 100 mL of normal saline, injected over 10 min. A 24-h continuous infusion of 3.5 mg/kg of naloxone diluted in 100 mL of saline.
Czlonkowske et al. (1988) [2]	Poland	Double- blind clinical trial	24	20	Patients that could receive treatment within 24 h of the ischemic infarct.	2 mL of saline followed by 0.4 mg of naloxone every 10 min for 30 min. In the control group, naloxone was replaced by saline.
Jabaily et al. (1984) [10]	United States	Single-blind clinical trial	13	Not given	Patients who suffered a stroke from 3–24 h.	2 mL of IV saline, then two or three ampules of naloxone from 0.8 to 1.2 mg at 10-min intervals.
Clark et al. (1999) [11]	United States	Double- blind, multicenter clinical trial	79 with 6 mg, 77 with 20 mg, and 81 with 60 mg.	75	Patients within 6 h of having an ischemic stroke.	Patients receive either 6 mg, 20 mg, or 60 mg of nalmefene over 24 h. 50 mL over a bolus in 15 min and 500 mL for 23.75 h.
Clark et al. (2000) [12]	United States	Double- blind, multicenter clinical trial	186	182	Patients within 6 h of having an ischemic stroke.	Patients receive 60 mg of nalmefene with a 10 mg bolis in 15 min. Later they receive 50 mg of infusion over 23.75 h. The control group received a placebo.
Li et al. (2017) [13]	China	Randomized controlled prospective clinical trial	120	116	Patients with symptoms within 3 days of the stroke.	10 days of nalmefene injection: 0.2 mg of nalmefene hydrochloride dissolved in 10 mL/0.9% sodium chloride.

mL: milliliters; mg: milligram; kg: kilogram; min: minutes.

**Table 2 brainsci-11-00805-t002:** Outcomes and conclusions of the clinical trials.

Author, Year	Drug	Outcome	Results of Treatment Group	Results of Control or Placebo Group	Main Conclusion
Fallis et al. (1984) [6]	Naloxone	NFS and B-endorphin levels (post-treatment)	NFS before naloxone: 34.4 ± 8.4 and 34.1 ± 8.9 post naloxone. B-endorphin levels: 9.6 ± 5.3	NFS before saline: 34 ± 8.2 and 34.6 ± 8.7 post saline. B-endorphin levels: 10.9 ± 9.8	Naloxone did not reverse neurological deficits. Plasma endorphin levels were not statistically significant.
Federico et al. (1991) [9]	Naloxone	CNE and BI	CNE: 5.6 ± 2; BI: 50.4 ± 11	CNE: 5.8 ± 2; BI: 49.5 ± 11	No statistical difference between both groups.
Czlonkowske et al. (1988) [2]	Naloxone	NSS	NSS: before naloxone: 61.50 ± 20 and after 2 weeks 75.46 ± 16.23	NSS: before saline: 76.65. ± 11.13 and after two weeks 82.10 ± 18.01	Highly statistical improvement *p* < 0.01. between treatment group and control.
Jabaily et al. (1984) [10]	Naloxone	Improvement of neurological deficits	3 out of 13 patients improved their neurological status.	No control group	3 of 13 patients improve their neurological deficits after 24 h.
Clark et al. (1999) [11]	Nalmefene	GOS + BI success, BI success, GOS success at three months	GOS + BI success (%): 84.8, 81.5, and 76.5 at 6 mg, 20 mg and 60 mg respectively.	GOS + BI success: 64.7	No significant difference was found. However, a clear tendency was seen in patients under 70 years.
			BI success (%): 88, 92,6, and 79.4 at 6 mg, 20 mg and 60 mg respectively.	BI success (%): 64.7	
			GOS success (%): 84.8, 81,5, 81,8 at 6 mg, 20 mg and 60 mg respectively.	GOS success (%): 67.6	
Clark et al. (2000) [12]	Nalmefene	GOS, BI, and NIHHS at three months	GOS + BI success: 66.9	GOS + BI success: 62.3	No significant difference was found. The tendency of patients under 70 years was again favorable, but again was not statically significant.
			BI success (%): 66.9	BI success (%): 64.1	
			GOS success (%): 68.1	GOS success (%): 62.9	
			NIHSS success (%): 36,2	NIHSS success (%): 32.9	
			Mortality (%): 16	Mortality (%):15.6	
Li et al. (2017) [13]	Nalmefene	NIHSS at 20 days, Glasgow Coma Scale at 0 and 10 days, matrix metalloproteinase-9 at 0.5 and 10 days, and magnetic resonance imaging perfusion at 0 and 10 days.	NIHSS: 22 ± 4 on day 0 and 17 ± 5 after 20 days	NIHSS 23 ± 4 on day 0 and 20 ± 5 after 20 days	In the treatment group, the NIHSS was decreased, the Glasgow Coma Scale was increased, matrix metalloproteinase-9 was decreased, and magnetic resonance imaging perfusion was increased compared to the treatment group with statistically significant results in all parameters.
			GOS: 7.2 ± 2.5 on day 0 and 9.5 ± 2.9 after 10 days	GOS: 7.1 ± 1.9 on day 0 and 8.1 ± 1.7 after 10 days	
			matrix metalloproteinase-9: 179 ± 76 at day 0 and 178 ± 56 at 10 days	matrix metalloproteinase-9: 189 ± 64 at day 0 and 210 ± 60 at 10 days	
			Cerebral blood flow mL/100 g/min: 43 ± 4 at day 0 67 ± 6 after 10 days	Cerebral blood flow mL/100 g/min: 40 ± 4 at day 0 59 ± 4 after 10 days	

NIHSS: (National Institutes of Healt Stroke Scale); NS: Neurological Score; GOS: Glasgow Outcome Score; CNE: Canadian Neurological Score; BI: Barthel Index; NSS: Neurological Status Score.

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
