# Peer review of "Opioid Antagonist in the Treatment of Ischemic Stroke"

_brainsci, 2021, doi:10.3390/brainsci11060805_

Round 1
Reviewer 1 Report
Ref: brainsci-1239584
Title: Opioid Antagonist in the Treatment of Ischemic Stroke
The subject of the paper is not new. There already exist a paper – Peyravian et al. 2019.
The paper is incomplete. I understand that the Authors wanted to point the clinical value of opioid antagonists. Nevertheless, there should be included the latest in vitro and in vivo and studies. For example, there are other papers which should be included e.g. Auttila et al. 2018; Sheng et al. 2018, Yang et al. 2011 and much much more.
Other comment:
- Materials and methods – Search Strategy - Was the search only in the context of naloxone? There is some inconsistency with figure 1 where is naloxone, naltrexone, nalmefene.
- Please correct the name of the Author – CzÅ‚onkowska not CzÅ‚onkowske
Reviewer 2 Report
The manuscript is well organized and potentially be interested. However, authors should discuss potential mechanisms of opioid antagonists in the treatment of stroke (both molecular and cellular), as well as animal models and future outlooks.
Author Response
Dear reviewer I re organized the document, and I include a section on animal and in vitro studies. and the section of future outlooks in the end discussed only the opioids mechanism, which gives the reader a better understanding of the topic. I include on the in vitro and animals studies section, papers of the last 10 years.
please see the sections 4.3 and 4.4
Round 2
Reviewer 1 Report
The authors responded to all my objections / doubts. After corrections made, I recognize the improvement of the manuscript.
Reviewer 2 Report
I do not have further comments.